# Predicting the Risk of Verticillium Wilt in Olive Orchards Using Fuzzy Logic

**Francisco Javier López-Escudero [1], Joaquín Romero [1], Rocío Bocanegra-Caro [1] and Antonio Santos-Rufo [1,2,*]**

[1]    Excellence Unit 'María de Maeztu' 2020-23, Department of Agronomy, Campus de Rabanales, University of Cordoba, 14071 Cordoba, Spain; ag2loesj@uco.es (F.J.L.-E.); joaquinromrod@gmail.com (J.R.); z82bocar@uco.es (R.B.-C.)

[2]    Department of Agroforestry Sciences, ETSI University of Huelva, 21007 Huelva, Spain

[*]    Correspondence: g02sarua@uco.es

**Abstract:** Developing models to understand disease dynamics and predict the risk of disease outbreaks to facilitate decision making is an integral component of plant disease management. However, these models have not yet been developed for one of the most damaging diseases in Mediterranean olive-growing areas (verticillium wilt (VW), caused by the fungus *Verticillium dahliae* Kleb.), although there are parameters (e.g., level of *V. dahliae* inoculum in the soil, level of susceptibility of the olive cultivar, isothermality, coefficient of variation of seasonal precipitation, etc.) that have previously been correlated with the severity of the disease. Using the data from previous VW studies conducted in the Guadalquivir Valley of Andalusia (one of the most damaged areas worldwide), in this work, a set of fuzzy logic (FL) models is developed with the aforementioned disease and climatic parameters, and the results are compared with machine learning (ML) models, of known effectiveness, to predict the risk levels of VW appearance in an olive orchard. Under these conditions, both groups of models were less effective than those previously studied with simpler models or models used under controlled conditions. However, the accuracy achieved with the most efficient FL model (60%; classification system based on fuzzy rules using the Ishibuchi method with a weighting factor) was somewhat greater than the efficiency achieved with the most efficient ML model (59.0%; decision tree classifier), in addition to being more appropriate (from a practical point of view) for the incorporation into a decision support system by allowing the risk of appearance of each observation to be known by providing rules for each of the combinations of the different parameters with similar precision. Therefore, in this study, we propose the FL methodology as suitable to act as an expert system for the future creation of a decision support system for VW in olives.

**Keywords:** disease risk prediction; fuzzy logic; inoculum density; isothermality; machine learning; resistance; *Verticillium dahliae*

## 1. Introduction

Verticillium wilt (VW), caused by the fungus *Verticillium dahliae* Kleb., represents the greatest phytopathological problem in olive cultivation as it causes yield losses and tree mortality worldwide [1–4]. This disease especially affects cultivated olive groves in Andalusia (southern Spain) [3], which is the world's leading olive tree grower, producing nearly half of the global olive oil production [5]. In this region, the spread of *V. dahliae* and the increase in the incidence of VW has been incessant from 1975 to the present [6,7], which has generated considerable concern in the olive sector, accentuated by the absence of effective control methods. VW should be managed according to an integrated management strategy that includes, as a preplanting (and the most effective) control measure, the quantification of the pathogen in soil while avoiding planting in infested soils, and the selection of cultivars resistant to or tolerant of *V. dahliae* [8,9]. After planting, the control measures mainly focus on avoiding an increase in the pathogen inoculum through chemical,

biological, physical, and cultural solutions [1,8,10–14]. However, such measures have not proved to be fully successful due to, among other reasons, the wide range of hosts *V. dahliae* presents, guaranteeing a constant expansion and multiplication in the soil and to its high persistence in the soil in its non-parasitic phase as microsclerotia—this pathogen has a parasitic phase in the host and a non-parasitic phase as microsclerotia, which can survive in field soils for up to 15 years [2,15]. This, among the widespread presence of the defoliating (D) pathotype of the pathogen in Andalusia, which is more virulent than the previously dominant non-defoliating pathotype (ND) (*V. dahliae* consists of two pathotypes with different degrees of virulence: D isolates cause defoliation in olives and cotton, whereas ND isolates cause wilting but no defoliation), has determined inoculum density in the soil (ID) as the main factor behind the spread of VW in olives [11,16].

On the other hand, it has also been firmly demonstrated that VW is conditioned by the host plant [8,17]. However, the susceptibility of the host plant is also related to the ID [8,11]. Thus, 'Picual' is extremely susceptible to *V. dahliae* when planted in soils with low inoculum densities (<1 propagules per gram of soil; ppg), while 'Arbequina' is moderately resistant to the pathogen under the same conditions. More than 250 olive cultivars have been evaluated and the main sources of resistance have exclusively been found in the cultivars 'Frantoio', 'Empeltre', and 'Changlot Real' [3,18].

Along with the ID and cultivar, the environment, broadly defined to include climate and preplanting factors that influence inoculum availability, determines the severity of VW epidemics. *V. dahliae* is a thermosensitive pathogen oscillating its optimum temperatures between 22–25 °C, similar to for most pathogenic species of *Verticillium* [19]. As for the disease, the severity of this is favored by temperatures > 20 and <25 °C in spring, followed by summers with mild temperatures, not exceeding 30–35 °C [10,20]. Recent work has shown the climatic factors that influence the occurrence of *V. dahliae* in Mediterranean olive groves in the Granada province (Andalusia, southern Spain) [21]. The authors tested a series of models with different combinations of climatic variables and demonstrated the influence of temperature on the increase in VW by stating that certain trends in thermal amplitudes generated this constant increase.

The interaction among various elements of the pathosystem indicate that VW progress in olives is likely to be complex. In this sense, a fundamental goal in botanical epidemiology is to predict the risk of disease at various spatio-temporal scales [22]. Thus, developing models to understand disease dynamics and predict such a risk of disease outbreaks to facilitate decision making is an integral component of plant disease management [23,24]. Among all the types of models used for this task, most of them are based on machine learning (ML). Within this type of methodology, the decision tree classifier and artificial neural networks are the models that are mainly used for herbaceous [25–28] or perennial crops, but by using images for automatically detected and recognized diseases [29] or a single set of parameters (i.e., disease or physiological parameters) under controlled conditions [30,31]. To efficiently predict the risk of disease at various spatio-temporal scales in the case of perennial crops, due to the complex interactions found, a more holistic view of the pathogen, using long-term data and not short periods of time with few locations, is needed. In this sense, fuzzy logic (FL) can handle the uncertainty of our knowledge and thus, it is especially useful when evaluating a system requires human experience that is expressed in 'linguistic' terms (e.g., a high risk of infection or low residual fungicide protection) [32]. As an example, this technique has been satisfactorily used to determine whether fungicide application is needed to control *Plasmopara viticola*, the causal agent of downy mildew, in a vineyard [33].

Even though the key elements of VW epidemiology are known, no tools are available for predicting the disease development in an orchard as influenced by the disease factors and environmental conditions. In this work, a set of FL models are developed by using disease and climatic parameters and compared with ML models of known effectiveness to predict the risk levels of the appearance of VW in a southern Mediterranean olive orchard. For that purpose, we use the dataset collected by [8,10–14] in the Guadalquivir Valley

of Andalusia. Model outputs can be used by technicians and farmers to define the risk of VW in a specific orchard and year, adapting crop management actions in the context of integrated pest management. The subsequent creation of a decision support system presents the utility of the generated models.

## 2. Materials and Methods

### 2.1. Data Collection and Parameters

In this study, we used the available data from previous studies conducted by the AGR-216 group from the University of Córdoba in the Guadalquivir Valley of Andalusia (Figure 1) during the last 30 years. These studies evaluated the effects of different techniques on the management of VW in olives, such as the resistance of cultivars [8], solarization [10], the level of inoculum in the soil [11], irrigation [12,13], biological control [14], and fertigation [34].

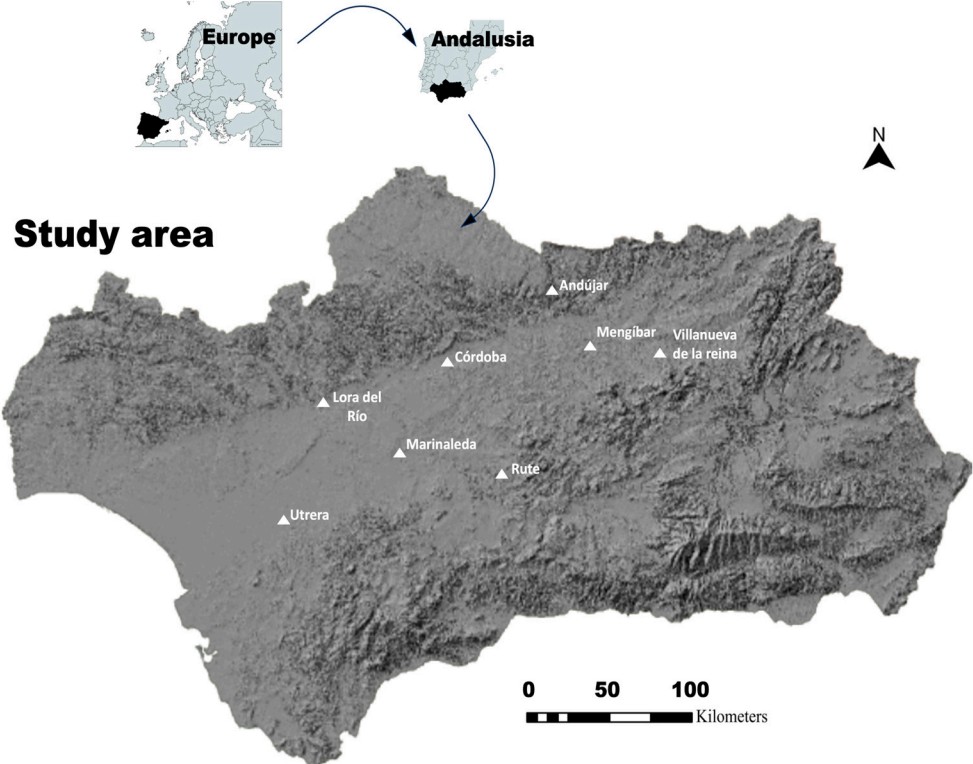

**Figure 1.** Location of studies conducted in the Guadalquivir Valley of Andalusia during the last 30 years by the AGR-216 group from the University of Córdoba, where the data serves to predict verticillium wilt risk in olives.

The location and years of the previously mentioned (six) studies can be seen in Figure 1 and Table 1. The protocols for the sampling, isolation, and identification of *V. dahliae* used to obtain the data, as well as the agronomic characteristics of fields, can be verified in the mentioned studies.

The data were exclusively collected from the control treatments of the experiments performed by [8,10–14]. Among all the disease parameters used, the disease incidence (DI) was selected as the most appropriate according to the proposed objectives. Similarly, the olive cultivar (Cv) and inoculum density of *V. dahliae* in the soil (ID) were selected as disease parameters (Table 1). In the case of the Cv, a numerical value was assigned according to the relative susceptibility index to VW (RSI) defined by [17] (Table 1). This index was 26.3% in the case of the extremely tolerant 'Frantoio' olive cultivar, 31.5% for 'Changlot Real' (tolerant), 75.8% for 'Arbequina' (moderately susceptible), 87.9% for 'Hojiblanca' (susceptible), and 100% for the extremely susceptible 'Picual' olive cultivar.

As climatic parameters, the monthly mean temperature (T) and accumulated precipitation (P) were selected. T and P were recorded by the meteorological stations near the trials and were used to calculate the isothermality (Isot) and coefficient of variation of seasonal precipitation (CvP), also as climatic parameters, respectively (Table 1). The isothermal parameter describes the oscillation of the temperature between day and night in relation to the oscillation between the summer and winter seasons, within the same year. The CvP, instead, is the quotient of the standard deviation of the total monthly precipitation with respect to the total monthly average precipitation [21]. The results of these parameters are shown as concrete indices, that is, without being multiplied by 100 (Table 1).

**Table 1.** Locations, years, and data range of the disease and climatic parameters employed to predict verticillium wilt risk in olives. Disease parameters were obtained from previous studies conducted in the Guadalquivir Valley of Andalusia by the AGR-216 group from the University of Córdoba.

| Locations/Years | Disease Parameters [1] | | | Climatic Parameters [2] | | | | References |
| | DI (%) | Cv (%) | ID (ppg) | T (°C) | P (mm) | Isot | CvP | |
| --- | --- | --- | --- | --- | --- | --- | --- | --- |
| Marinaleda/1994–1997 | 40.3–87.7 | 87.9 | 14.1–28.6 | 4.1–29.4 | 0–68.8 | 0.1–0.8 | 1–3 | |
| Lora del Río/1994–1997 | 39.3–95.3 | 100 | 0–13.3 | 8.3–37.1 | 0–16.1 | 0.1–1.1 | 1–1.6 | [10,34] |
| Rute/1994–1997 | 8–34 | 100 | 0.5–3.6 | 8–27.4 | 0–21.4 | −1.8–0.8 | 1–1.9 | |
| Córdoba/2015–2016 | 0–100 | 100 | 5.5–10.3 | 8.8–30.6 | 0–3.4 | 0.1–0.6 | 0.9–1 | [34] |
| Córdoba/2011–2015 | 0–100 | 26.4–100 | 1.2–9.8 | 5–28.5 | 0–13.2 | 0–0.5 | 0.8–1.4 | [12,13] |
| Andújar/2011–2015 | 0–91.3 | 100 | 5.4–5.4 | 6.1–28.8 | 0–8.6 | 0.1–0.5 | 0.8–1.1 | |
| Córdoba/2001–2003 | 0–66.6 | 100 | 0–10 | 8–28.1 | 0–4.8 | 0.2–0.6 | 0.9–1.1 | [11] |
| Utrera/2010–2011 | 4.0–96.0 | 26.4–100 | 21–21 | 10.8–22.5 | 0.6–6.8 | 0.1–0.7 | 0.9–0.9 | [8] |
| Villanueva de la Reina/2015–2018 | 0–93.3 | 26.4–100 | 0–80 | 5.8–28.1 | 0–4.8 | 0.1–0.7 | 0.8–1.2 | [14] |

[1] The disease incidence (DI), olive cultivar (Cv), and inoculum density of *V. dahliae* in the soil (ID; propagules per gram of soil) were selected as disease parameters. In the case of the Cv, a numerical value was assigned according to the relative susceptibility index to VW (RSI) defined by [17]. RSI values were 26.3% in the case of the 'Frantoio' olive cultivar, 75.8% for 'Arbequina', 87.9% for 'Hojiblanca', and 100% for 'Picual'. The protocols for the sampling, isolation, and identification of *V. dahliae* used to obtain these data can be checked in the indicated studies. [2] Monthly mean temperature (T), accumulated precipitation (P), isothermality (Isot), and the coefficient of variation of seasonal precipitation (CvP) were selected as climatic parameters. The isothermal parameter describes the oscillation of temperature between day and night in relation to the oscillation between the summer and winter seasons, within the same year. The CvP, instead, is the quotient of the standard deviation of the total monthly precipitation with respect to the total monthly average precipitation [21]. The results of these parameters are shown as concrete indices, that is, without being multiplied by 100.

Finally, the dataset used consisted of 2366 data with a data range of 0–100% for DI, 26.4 to 100.0% for Cv, 0.0 to 80.0 for ID, 4.1–37.2 for T, 0.0–68.8 for P, −1.8–1.1 for Isot, and 0.8–3.0 for CvP (Table 1).

### 2.2. Modeling Approach and Software

The general response of the experimental measures of the parameters DI, Cv, ID, T, P, Iso, and CvP was first explored by means of a cluster analysis. The purpose of this analysis was to establish functional groups of correlated experimental measures and thus assign each subset of measures a class or level of disease risk.

Then, the relationships between these classes and the studied parameters (DI, Cv, ID, T, P, Iso, and CvP) were evaluated using FL models. Among all the learning models applied to this methodology, in this work, those algorithms that focused on data classification tasks were used. These algorithms were the fuzzy rule-based classification system using the CHI method (FRBCS.CHI; [35]), classification system based on fuzzy rules using the Ishibuchi method with the weight factor (FRBCS.W; [36]), genetic fuzzy system based on genetic cooperative competitive learning (GFS.GCCL; [37]), fuzzy hybrid of genetics-based machine learning (FH.GBML; [36]), and structural learning algorithm in a vague environment (SLAVE; [38]). A 10-fold cross-validation was considered to validate these FL models.

Supervised ML algorithms were also fitted to the parameters as explanatory parameters, and the DI or disease risk class was the response parameter, taking the lowest disease risk class as the reference category. The dataset was split 4:1 into training and test sets, respectively, and a 10-fold cross-validation was used to estimate the accuracy of the models, with accuracy as the parameter for scoring. The best learned classifier in the test set was then assessed and used to identify groups that were homogeneous in terms of disease risk. The results are summarized as a final accuracy score, a confusion matrix, and a classification report, which are common performance measures for machine learning classification of severity classes in various plant diseases [26,31].

A cluster analysis and FL implementation were performed in the R software environment [39] using the version 2023.03.0+386 of RStudio [40] with cluster [41] and frbs [42] packages, respectively. The ML algorithms were implemented simultaneously using the Lazy predict library [43] in version 0.19.1 of the Python software [44].

## 3. Results

### 3.1. Clustering

The hierarchical cluster analysis yielded five functional groups (A–E) among the 2366 combinations (Figure 2). Group A consisted of 183 combinations with a very high DI level and was associated with a very high ID, a moderate to high Isot, a moderately susceptible Cv, and a low CvP. In the opposite case, with a very low DI level (group E), 1352 combinations were included, with Cv being predominantly tolerant; the ID with low ppg values and the Isot and CvP were positioned with values that were medium to low. The rest of the groups (B, C, and D) included 341, 231, and 259 combinations with severe, moderate, and low levels of DI (severe, moderate, and low risks), respectively (Figure 2). As a result of this analysis, five classes were defined as risk levels of VWO occurrence: very severe, severe, moderate, low, and very low.

### 3.2. Fuzzy Logic (FL)

Between all the FL models employed, FRBCS.W was the best that described and predicted the VW risk classes with an error of 40.0% and accuracy of 60.0% (Table 2). The error and accuracy of the genetic algorithm GFS.GCCL was the highest and lowest of all the algorithms used (70.0 and 20%, respectively). The results of the other algorithms can be observed in Table 2.

**Table 2.** Accuracy of the fuzzy logic classification algorithms executed to discriminate among verticillium wilt risk classes based on several disease and climatic parameters.

| Algorithms | Accuracy (Error) |
|---|---|
| Fuzzy Rule-Based Classification System using the CHI method (FRBCS.CHI) | 0.50 (0.55) |
| Fuzzy Rule-Based Classification System using the Ishibuchi method with the weight factor (FRBCS.W) | 0.60 (0.40) |
| Genetic Fuzzy System based on Genetic Cooperative Competitive Learning (GFS.GCCL) | 0.20 (0.70) |
| Fuzzy Hybrid of Genetics Based Machine Learning (FH.GBML) | 0.50 (0.46) |
| Structural Learning Algorithm in a Vague Environment (SLAVE) | 0.40 (0.60) |

Based on the membership function from the FRBCS.W, some of the *if-then* rules with high success factors are shown in Table 3 (35 rules out of a total of 205).

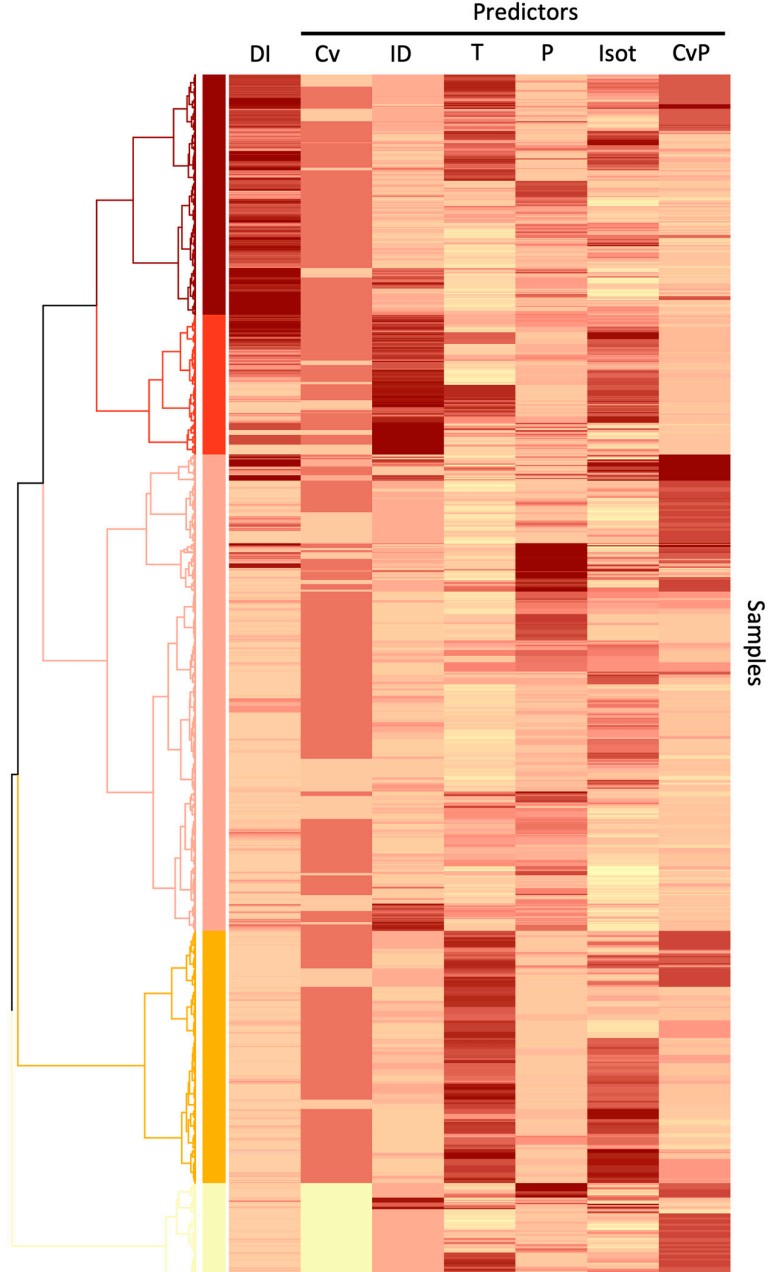

**Figure 2.** Dendrogram showing the results of the cluster analyses and a heat map representation of disease and climatic parameters in a study where the risk of verticillium wilt in olives was predicted using the fuzzy logic model. The seven parameters selected for the heat map representation were related to verticillium wilt (three parameters: final disease incidence (DI), the olive cultivar (Cv; RSI), and the inoculum density of *Verticillium dahliae* in the soil (ID)) and climate (four parameters: the monthly mean temperature (T; °C) and precipitation (P, mm), the temperature oscillation between day and night in relation to the oscillation between the summer and winter seasons, within the same year (isothermality; Isot), and the coefficient of variation of seasonal precipitation (CvP)). Agglomerative cluster analyses were performed based on the Spearman's correlation matrix calculated from values of the different parameters using the Ward method. Cluster groupings of samples represented in different colors were estimated on the basis of the average silhouette width according to the Mantel statistic. In the heat map, for each column, the cells represent the relative values of each parameter for each sample from six previous studies (nine experiments with 2366 data in total) conducted in the Guadalquivir Valley of Andalusia during the last 30 years by the AGR-216 group from the University of Cordoba.

**Table 3.** Fuzzy *if-then* rules to discriminate among verticillium wilt risk classes based on disease and climatic parameters. The fuzzy control system employed was the classification system based on fuzzy rules using the Ischibuchi method with the weight factor (FRBCS.W; see Section 2.2 for details).

| Rules No. | Rules [1,2] | | | | | | Risk | Success Factor (%) |
| | Cv (%) | ID (ppg) | T (°C) | P (mm) | Isot | CvP | | |
|---|---|---|---|---|---|---|---|---|
| 1 | Mod. susceptible | Very low | Low | Very low | High | Very low | Very low | 38.0 |
| 2 | Tolerant | Very low | Low | Very low | High | Very low | Low | 23.0 |
| 3 | Tolerant | Very low | Medium | Very low | High | Very low | Very low | 39.0 |
| 4 | Tolerant | Very low | High | Very low | High | Very low | Very low | 39.0 |
| 5 | Tolerant | Very low | Low | Very low | High | Very low | Severe | 22.0 |
| 6 | Tolerant | Very low | Very low | Very low | High | Very low | Moderate | 39.0 |
| 7 | Tolerant | Very low | Low | Very low | High | Very low | Very low | 39.0 |
| 8 | Tolerant | Very low | Very low | Very low | High | Very low | Low | 23.0 |
| 9 | Tolerant | Very low | Low | Very low | High | Very low | Severe | 22.0 |
| 10 | Tolerant | Very low | Low | Very low | High | Very low | Very severe | 23.0 |
| 11 | Tolerant | Very low | Low | Very low | High | Very low | Moderate | 18.0 |
| 12 | Tolerant | High | Very low | Very low | High | Very low | Severe | 22.0 |
| 13 | Tolerant | Very low | Very low | Very low | High | Very low | Low | 23.0 |
| 14 | Tolerant | Very low | High | Very low | High | Very low | Very low | 39.0 |
| 15 | Mod. susceptible | Very low | High | Very low | High | Very low | Very low | 39.0 |
| 16 | Tolerant | Very low | Very low | Very low | High | Very low | Low | 23.0 |
| 17 | Mod. susceptible | Very low | Very low | Very low | High | Very low | Very low | 39.0 |
| 18 | Tolerant | Very low | High | Very low | Very high | Very low | Very severe | 23.0 |
| 19 | Tolerant | Very low | High | Very low | High | Very low | Severe | 22.0 |
| 20 | Tolerant | Very low | High | Very low | High | Very low | Moderate | 22.0 |
| 21 | Mod. susceptible | High | Low | Very low | High | Very low | Very low | 18.0 |
| 22 | Tolerant | Very low | Very low | Very low | Medium | Very low | Low | 23.0 |
| 23 | Tolerant | Very low | Medium | Very low | Very low | Very low | Very severe | 23.0 |
| 24 | Tolerant | Very low | Medium | Very low | Very low | Very low | Moderate | 18.0 |
| 25 | Tolerant | Very low | Medium | Very low | Very low | Very low | Severe | 22.0 |
| 26 | Mod. susceptible | Very low | Very low | Very low | Very low | Very low | Low | 23.0 |
| 27 | Extrem. tolerant | Very low | Very low | Very low | Very low | Very low | Very low | 39.0 |
| 28 | Extrem. tolerant | Very low | Very low | Very low | Very low | Very low | Low | 23.0 |
| 29 | Tolerant | Very low | Medium | Very low | Very low | Low | Very severe | 23.0 |
| 30 | Mod. susceptible | High | Very low | Very low | Very low | Very low | Very low | 39.0 |
| 31 | Tolerant | High | Very low | Very low | Very low | Very low | Severe | 22.0 |
| 32 | Tolerant | Very low | High | Very low | Very high | Very low | Very low | 39.0 |
| 33 | Tolerant | Very low | High | Very low | Very high | Very low | Low | 23.0 |
| 34 | Mod. susceptible | Very low | Medium | Very low | Very low | Very low | Very low | 39.0 |
| 35 | Tolerant | High | Medium | Very low | Very low | Very low | Severe | 22.0 |

[1] As an example, the fuzzy logic terms for rule 1 should be as follows: *If* (Cv is Mod. Susceptible) *and* (ID is Very low) *and* (T is Low) *and* (P is Very low) *and* (Isot is High) *and* (CvP is Very low) *then* (VW risk is Very low)
[2] Two disease and four climatic parameters were used: the level of susceptibility of the olive cultivar (Cv; RSI), the inoculum density of *Verticillium dahliae* in the soil (ID), the monthly mean temperature (T) and precipitation (P), the isothermality (Isot), or the temperature oscillation between day and night in relation to the oscillation between the summer and winter seasons within the same year (Isot), and the coefficient of variation of seasonal precipitation (CvP).

Furthermore, the plot of membership functions when the FRBCS.W method was employed can be seen in Figure 3. It shows that there are six input attributes that have five linguistic terms (trapezoid) for each attribute. The normalized range of the data for each parameter and the degree of the membership function are presented in the horizontal and vertical axes, respectively.

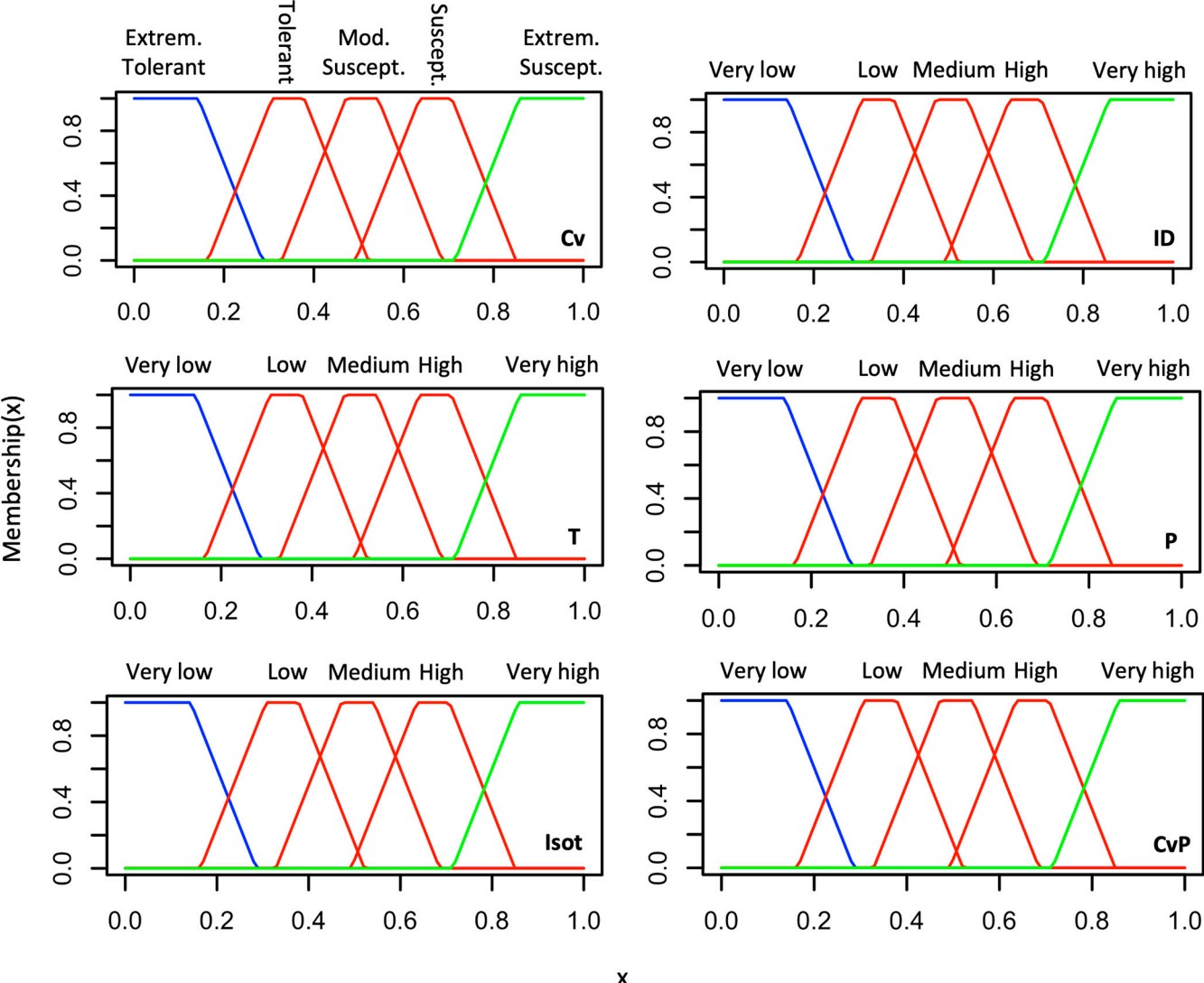

**Figure 3.** Normalized input membership functions for VW risk probability using different disease and climatic conditions. Two disease and four climatic parameters were used: the level of susceptibility of the olive cultivar (Cv; RSI), the inoculum density of *Verticillium dahliae* in the soil (ID; ppg), the monthly mean temperature (T; °C) and precipitation (P, mm), the isothermality, or the temperature oscillation between day and night in relation to the oscillation between the summer and winter seasons within the same year (Isot), and the coefficient of variation of seasonal precipitation (CvP). The fuzzy control system employed was the classification system based on fuzzy rules using the Ischibuchi method with the weight factor (FRBCS.W; see Section 2.2 for details).

As an example, for the Cv parameter, the subset below the blue line represented all the data belonging to extremely tolerant cultivars (rules 27 and 28; Table 3), those below the first red line corresponded to data for tolerant cultivars (rules 2, 4–14, 16, 18–20, 22–25, 29, 31–33, and 35; Table 3), and the subsequent trapezoids represented those that were considered as moderately susceptible (rules 1, 15, 17, 21, 26, 30, and 34; Table 3), etc. Each trapezoid was determined by four values that corresponded to its four vertices, whose values were equivalent to its x coordinate on the graph's axis, being 0.00, 0.14, and 0.28 for the subsets of the data corresponding to the extremely tolerant cultivars; 0.16, 0.31, 0.38, and 0.52 for tolerant cultivars; 0.33, 0.49, 0.54, and 0.68 along with 0.47, 0.63, 0.71, and 0.85 (subsequent trapezoid) for moderately susceptible and susceptible cultivars, respectively; and, 0.71, 0.86, and 1.00 for extremely susceptible cultivars. As can be seen, since these are normalized data, the graphs of the six parameters considered in the study follow the

same shape. This means that everyone in the model has the same influence when it comes to providing the result, although the ranges of the values for each of the variables are completely different (e.g., an RSI value of 26.35% can be in the same range as an ID of 0.2 ppg). Moreover, the areas of overlap that occurred between the trapezoids could result in a value on the x axis belonging to two different trapezoids that were equivalent to two different data regions (for example, moderately susceptible and tolerant in the case of the Cv parameter). This occurred due to the fact that, for the same rule, the success factor was different. Therefore, it was the logic we applied based on the knowledge we already had about this disease that determined, along with its success factor, its validity.

*3.3. Machine Learning (ML)*

The results obtained for each of the ML models employed are shown in Supplementary Table S1. Among them, the decision tree classifier was the algorithm that provided the best results in terms of the accuracy and F1 value and, therefore, it was validated with the test set for a final independent verification of the accuracy of the model. With the test sets, this algorithm showed precision values of 92.0%, 56.0%, 48.0%, 59.0%, and 39.0% for the very low, low, moderate, severe, and very severe classes, respectively, with an average F1 value of 59.0% (Table 4).

**Table 4.** Classification report of a decision tree classifier directly run on the test set to discriminate among verticillium wilt risk classes based on several disease and climatic parameters.

|  | Precision | Recall | F1-Score | Support |
|---|---|---|---|---|
| Very low [1] | 0.92 | 0.91 | 0.92 | 317 |
| Low | 0.56 | 0.64 | 0.60 | 53 |
| Moderate | 0.48 | 0.57 | 0.52 | 54 |
| Severe | 0.59 | 0.51 | 0.55 | 81 |
| Very severe | 0.39 | 0.38 | 0.38 | 40 |
| avg/total | 0.59 | 0.60 | 0.59 | 545 |

[1] Risk of verticillium wilt occurrence in an olive orchard.

Moreover, the thresholds of the ID, Cv, T, P, Isot, and CvP parameters that discriminated between VW risk classes were determined by using the XGBoost library for Python software [45]. The pruned tree generated by the classification tree contained five parameters (ID, Cv, T, Isot, and CvP) with 13 terminal nodes (Figure 4). The ID parameter was the main factor (i.e., the first division parameter) that differentiated between samples in the very severe VW risk class from those samples belonging to the severe, moderate, low, and very low classes, with a relative ID threshold of 13.15 ppg (Figure 4).

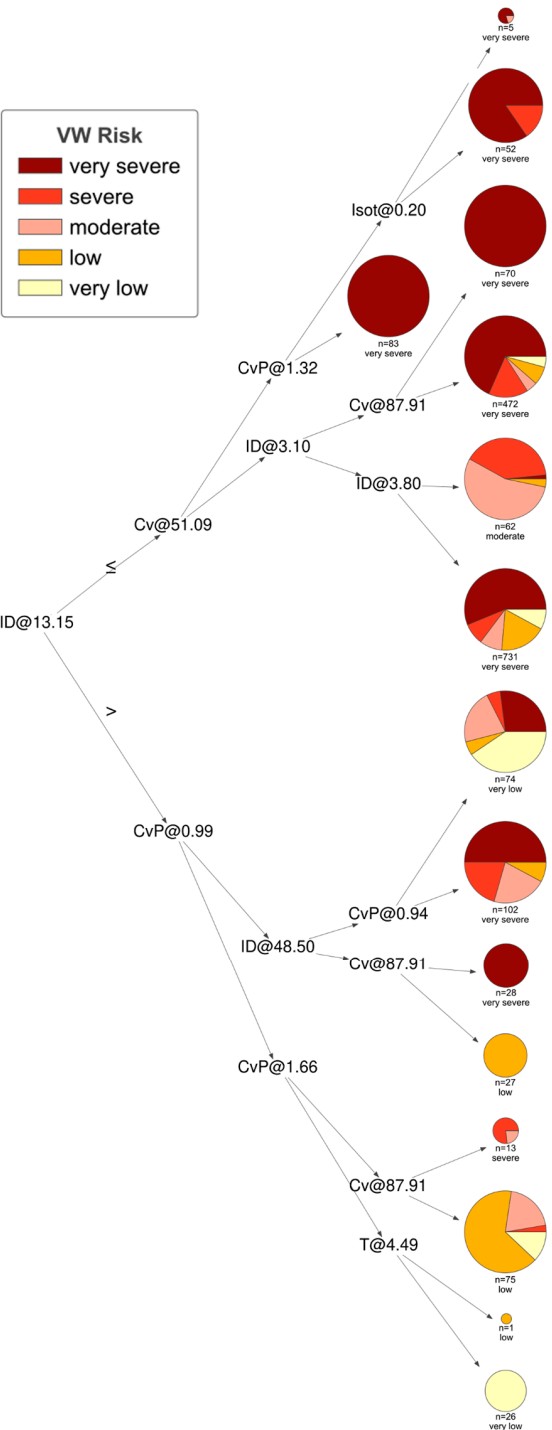

**Figure 4.** Classification tree to discriminate among verticillium wilt risk classes based on disease and climatic parameters. Two disease and four climatic parameters were used: the level of susceptibility of the olive cultivar (Cv; RSI), the inoculum density of *Verticillium dahliae* in the soil (ID; ppg), the monthly mean temperature (T; °C) and precipitation (P, mm), the isothermality, or the temperature oscillation between day and night in relation to the oscillation between the summer and winter seasons within the same year (Isot), and the coefficient of variation of seasonal precipitation (CvP). Risk class indicates the severity of verticillium wilt symptoms from very low to very severe symptom developments. The histogram for each terminal node represents the percentage of samples in each severity class. The data include a training set of 473 samples selected at random from a total set of 2366 samples from six previous studies (eight experiments in total) conducted in the Guadalquivir Valley of Andalusia during the last 30 years by the AGR-216 group from the University of Cordoba.

## 4. Discussion

There are many advantages to predicting the risk of suffering from VW in an olive orchard. Among them, one of the most important is the possibility of avoiding planting in orchards where the conditions indicate a high risk of VW. Likewise, knowing the moment at which said risk increases could facilitate the application of preventive control measures. However, what is known to date about VW prediction has not been adequate enough to obtain reliable models for olives, although there are parameters (e.g., level of *V. dahliae* inoculum in the soil, susceptibility level of the olive cultivar, isothermality, etc.) that have been previously correlated with the severity of the disease. In this study, these parameters were used to develop VW risk assessment models by using the long-term data collected from previous studies across diverse fields in the Guadalquivir Valley in Andalusia. Such a risk was assessed by FL algorithms and the results were compared with ML algorithms with a known high efficiency. Under these conditions, the accuracy of the most efficient FL model (FRBS.W) turned out to be somewhat higher than the most efficient ML model (decision tree classifier). Moreover, the FL model was the most adequate for allowing us to understand the risk of appearance for each observation providing rules for each of the combinations of the different parameters with a similar accuracy. Thus, it might be adequate to function as an expert system for the future creation of a decision support system for VW in olives.

According to the result obtained by the cluster analysis and the conclusions obtained by [21], the isothermality is proportional to the risk of occurrence of VW in olives. The combinations where this parameter obtained a higher value was when the risk of VW occurrence was more severe. This was consistent with the results obtained in other studies where it was observed that this disease was favorable to climatic fluctuations, in this case temperature, rather than by an average temperature during a specific day [4,46]. It is known that the optimum temperature for *V. dahliae* is in the range of 22–25 °C; therefore, it was assumed that the high temperatures reached in summer suppressed the development of the pathogen in the main Mediterranean olive areas [47,48].

As expected, the FL models used were effective in predicting VW risk when implemented with the selected parameters of this study (Cv, ID, T, P, Isot, and CvP). This technique was proposed by [49] with the aim of representing the knowledge of experts in a set of fuzzy *if-then* rules to solve complex real-life problems. It is characterized by being able to handle the uncertainty of our knowledge, allowing us to achieve the best possible reasoning [50]. Thus, the FL model is currently applied in many scientific areas [51], including plant pathology. In this area, this technic has been mainly used to develop expert systems that help researchers, advisors, and farmers identify the pathogens associated with observed symptoms [52]. Therefore, we can consider that the FL model has an intuitive technology as a predictive model, is easy to transfer to farmers or field technicians, and, most importantly, is suitable as the connection for a decision support system. The most effective FL technique in this study (FRBCS.W) was developed in a similar way to that developed by [33]. To perform this, first, a set of rules was developed resulting from a combination between the different sublevels of each of the parameters considered in the study. Subsequently, a result was obtained: a risk of VW development in this study or the decision of whether to apply a fungicide to *Plasmopara viticola* in a vineyard in the study by [33], obtaining the precision of the comparison between the results that the predictive model provided and what an expert would do in the same case. However, the precision levels of such an algorithm in this work were not very high (60.0%) compared to those found in the previously mentioned work (99.2%; [33]). The lower precision value obtained in this study could have been due to the number of classes that were considered (five: very low, low, medium, high, and very high VW risks), while [33] only used two unique solutions or classes: applying or not applying a fungicide. In general, the varied origin of the data used in this study (field, microplots, etc.), among other aspects, could reduce the accuracy of these models. It was possible that the behavior of the same cultivar or the response of the plant to the same ID for the different trials was different and that this

generated certain inconsistencies, both in the cluster analysis and the execution of the model. If the results obtained were compared with the decisions that a real VW expert would have made regarding the same combination of variables, the results could have been different. In fact, according to Table 3, although most of the rules do not have a relatively high success value, it makes sense to be guided by the theories it establishes if we focus on the combination of the different parameters used. The risk conferred because of this combination was equivalent to what really occurred in the development of the disease, and therefore these theories would be quite useful to use for the prediction of VW for practical purposes. Apart from these rules, this model graphically described each of the variables (Figure 3). This would allow us to visually perceive the range of values where the new data for each parameter would be located. By combining the linguistic terms (susceptible cultivar, high ID, medium Isot, etc.), we could create rules that, when compared with the results in Table 3, would present the risks existing in a specific olive orchard.

Among the ML algorithms evaluated in this study, the decision tree classifier yielded the highest classification accuracy, and when it was used to determine the thresholds of the different parameters that discriminated between the VW risk levels, the inoculum density parameter was the best indicator (Figure 4). This algorithm was also used to determine thresholds that discriminated between classes of VW severity for disease (e.g., density of micropropagules [31]) or physiological (e.g., ethylene production [30]) parameters. In these studies, higher accuracy results (73.0–75.0%) were achieved compared to those obtained in this study (59.0%; Table 4). As previously mentioned, the varied origins of the data used in this study (field, microplots, etc.), among other aspects, could also reduce the accuracy of these models. However, in both decision trees used in this study, and in [31], the inoculum density parameter was the best indicator for the detection of VW, as was previously demonstrated [11]. However, according to the results obtained in this work, although the threshold value of 13.2 ppg for *V. dahliae* in the soil is not exceeded, in the second level, those plantations with cultivars with an RSI > 51.1% (e.g., Arbequina, Hojiblanca or Picual) are at moderate to very severe risks of the appearance of VW, regardless of the values of the rest of the parameters (Figure 4). However, if that ID threshold is exceeded, or it remains the same, it is the coefficient of variation of precipitation (CvP) that becomes more important in predicting what will occur. Depending on whether this parameter exceeds the threshold of 1.7, we would have a very low to severe risk of disease onset. However, if the inoculum density increases until it reaches the threshold level of 48.5 ppg, we would have a very severe risk. These results reaffirm the importance of the ID parameter in the development of VW.

Under the conditions of this study, the FL method was more adequate than the ML method for better adjusting to the uncertainties presented by VW in olives. Although the accuracy achieved with the FL method turned out to be somewhat higher that the accuracy achieved with ML, this technique was especially useful for the evaluation of the selected parameters, which required human knowledge expressed in linguistic terms to reach a conclusion regarding the subsequent creation of decision support systems. Thus, the FL method was adequate for the subsequent creation of decision support systems, which can help olive farmers in the process of decision making for VW.

## 5. Conclusions

There are disease and climatic parameters that influence the infection of olive trees by *V. dahliae* and the development of VW. However, the joint influence of both groups of parameters on this disease is unknown. In this study, 2366 data of disease (inoculum density and susceptibility of olive cultivar) and climatic (temperature, precipitation, isothermality, and coefficient of variation of seasonal precipitation) parameters were used to predict the risk of appearance of VW through the FL technique, and the results were compared with ML algorithms of known high efficiency. The data were collected from previous studies and corresponded to eight *V. dahliae* naturally infested olive orchards of different natures and geographical locations. Among several FL and ML algorithms, the classification system

based on fuzzy rules using the Ishibuchi method with the weight factor (FRBCS.W) and the decision tree classifier were the algorithms with higher accuracies (60.0 and 59.0%, respectively). Although the accuracy of the FRBCS.W was almost similar to that observed for the decision tree classifier, the FL algorithm turned out to be more intuitive and easier to understand than the ML method, and thus, it could be useful for the prediction of VW for practical purposes. In addition to providing rules for each of the combinations of the different parameters, this model graphically described these parameters, allowing us to visually perceive the range of values where the new data for each parameter would be located. Based on the simple information provided by the farmer, the model could provide a value (risk level) and reason (rule) that would allow the user to understand what was occurring in the olive orchard and anticipate the disease. These techniques, along with other newer network models for a potentially better detection accuracy, must be validated with real data from commercial farms in future works with the aim of increasing their levels of precision. In this way, if the data were exclusively obtained from field plots, the number of inconsistencies would be reduced, the similarities between the data would be greater, and the validity of the model would be higher. The following steps would be the construction of a decision support system, which is a system that provides information of interest in a concise manner, serving as a guide for the olive farmer to make decisions based on the collected data.

**Supplementary Materials:** The following supporting information can be downloaded at: https://www.mdpi.com/article/10.3390/agriculture13112136/s1, Table S1: Result of machine learning algorithms implemented with the Lazy Predict library in Python software to discriminate among verticillium wilt risk classes based on the following disease and climatic parameters: the level of susceptibility of the olive cultivar (Cv; RSI), the inoculum density of *Verticillium dahliae* in the soil (ID; ppg), the monthly mean temperature (T; °C) and precipitation (P, mm), the isothermality, or the temperature oscillation between day and night in relation to the oscillation between the summer and winter seasons within the same year (Isot), and the coefficient of variation of seasonal precipitation (CvP).

**Author Contributions:** Conceptualization, F.J.L.-E., J.R. and A.S.-R.; data curation, R.B.-C and F.J.L.-E.; formal analysis, R.B.-C. and A.S.-R.; funding acquisition, F.J.L.-E.; investigation, F.J.L.-E. and A.S.-R.; methodology, A.S.-R. and J.R.; project administration, F.J.L.-E.; resources, F.J.L.-E.; validation, A.S.-R.; writing—original draft, F.J.L.-E., R.B.-C. and A.S.-R.; writing—review and editing, A.S.-R. and F.J.L.-E. All authors have read and agreed to the published version of the manuscript.

**Funding:** This research was supported by the Seresco S.L. Company within the project CIP-OLIVE (cloud-based integrated platform for monitoring pests; www.modem-ivm.eu (accessed on 8 August 2023)) partly funded by the Spanish Centre for Industrial Technological Development (CDTI).

**Institutional Review Board Statement:** Not applicable.

**Data Availability Statement:** The data are contained within the article.

**Acknowledgments:** We especially wish to thank the Agroforestry Pathology group from the University of Cordoba (AGR-216) for permitting the authors to collect the data for this study.

**Conflicts of Interest:** The authors declare no competing financial interest.

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
