# Peer review of "Predicting the Risk of Verticillium Wilt in Olive Orchards Using Fuzzy Logic"

_agriculture, doi:10.3390/agriculture13112136_

Round 1

Reviewer 1 Report

Comments and Suggestions for Authors

The authors propose a set of fuzzy logic models for olive Verticillium wilt (VW) prediction. The total amount of work involved from data set production to modeling to final analysis is sufficient. The overall quality of the manuscript is reasonable. I list below some comments to improve the presentation. 

The Introduction spent too much describing the pathogenesis and principles of VW. Introduction is recommended to include following sections: 1. Adverse impact of VW to olive growing industry; 2 Literature review on machine learning in pest and disease prevention (who,  what was done, what are the contributions or limitations); 3 Key concerns of this study; 4 Main work.  

Line169-182: All URLs in the text should be in the form of references 

As an important means of argumentation of this manuscript, the supplementary should be made into visual images or put directly into results, thus arguing that the author's method is superior to existing models. 

In Results, the authors should add a confusion matrix analysis of the results of the FL, and then analyze them by selecting a few well-performing models in Table s1 for comparison. 

Line 280: figure1? figure4? 

In the discussion section, the author presented a long comparative analysis and argues the superiority of the proposed model, however, it lacks a concise paragraph to state the main contribution of the whole manuscript. 

The Conclusion is also a bit wordy. The future work could be brought up in a more concise way.

Reviewer 2 Report

Comments and Suggestions for Authors

The text discusses the issue of diseases in olive cultivation and proposes a fuzzy logic model based on previous experimental data to improve the accuracy of disease detection. The article analyzes various indicators for evaluation and compares the results with machine learning models, effectively addressing the identification problem of Verticillium wilt in olive groves. It demonstrates the advantages and feasibility of the selected fuzzy logic algorithm. However, the text has lower accuracy in disease prediction. It is recommended to try using newer network models for potentially better detection accuracy and to provide more details on the compared models, possibly through the addition of charts and illustrations. Furthermore, there are multiple errors in the expression of details in the text, such as inconsistent formatting and layout errors.

Shortcomings and suggestions:

1.     Line 280, Figure 1 should be Figure 4.

2.     Figure 4 mentioned in lines 275, 278, 360, and 372 could not be found.

3.     In lines 108 to 110 of the second chapter, it is recommended to list the content in the order of references or to change the order of the references.

4.     In the third chapter, the explanation of the machine learning experiments section is insufficient. It is suggested to include specific accuracy comparison charts to provide additional clarification.

5.     The accuracy obtained in the manuscript for the experiments is only 59.0%. It is recommended to conduct more experiments and innovations to improve the model accuracy.

Reviewer 3 Report

Comments and Suggestions for Authors

In the paper “Predicting risk of Verticillium wilt in olive using fuzzy logic”, authors developed a set of fuzzy logic models by using disease and climatic parameters, and compared with machine learning models of known effectiveness, to predict the risk levels of the appearance of Verticillium Wilt in southern Mediterranean olive orchard.  The article is interesting and represents a further step forward on this issue. The work is clearly described, and tables and figures are also well detailed.

I appreciated the article and pleasantly verified great attention to details such as in the figures, tables, and correct use of acronyms (otherwise I am accustomed to pointing out numerous though minor corrections on these aspects).

The introduction is clear, frames the topic well, cites previous work and clearly states what the objectives are. The topic is not very original as other attempts have been made in the past on the classification of Verticillium damage and prediction. In any case, considering the importance of olive cultivation in Mediterranean countries, it will certainly be interesting for potential readers to read a paper on a further study that should hopefully be completed by the authors with a subsequent in-depth study (aiming to obtain a model with higher accuracy). I appreciated the extensive section on Discussion of results that shows the comparison with other similar works dwelling in particular on the accuracy achieved by the algorithm and the advantages of using FL in this case and in a decision support system which should serve as a guide for the olive farmer. Finally, the Conclusions section summarizes the content of the work well and indicates the authors' future intentions. The references are many and appropriate. 
